# The Characterization Analysis of the Oil-Immersed Transformers Obtained by Area Elimination Method Design



**Nuttee Thungsuk [1], Narong Mungkung [2], Thaweesak Tanaram [3], Arckarakit Chaithanakulwat [1], Somchai Arunrungsusmi [2,*], Wittawat Poonthong [2], Apidat Songruk [2], Khanchai Tunlasakun [2], Chalathip Chunkul [2], Tanes Tanitteerapan [2], Toshifumi Yuji [4] and Hiroyuki Kinoshita [5]**

[1] Department of Electrical Engineering, Dhonburi Rajabhat University Samut-Prakan, Samut-Prakan 10540, Thailand; nuttee.t@dru.ac.th (N.T.); chaithanakul@gmail.com (A.C.)
[2] Department of Electrical Technology Education, King's Mongkut's University of Technology, 126 Prachauthit Road, Bangmod Thungkru, Bangkok 10140, Thailand; narong_kmutt@hotmail.com (N.M.); poonthong_golf2538@outlook.com (W.P.); songruk.apidat38@hotmail.com (A.S.); khanchai.tun@kmutt.ac.th (K.T.); chalathip.chu@kmutt.ac.th (C.C.); tanes_kmutt@yahoo.com (T.T.)
[3] Faculty of Industrial Technology, Pibulsongkram Rajabhat University, Phitsanulok 65000, Thailand; t.tanaram.t@psru.ac.th
[4] Faculty of Education, University of Miyazaki, Miyazaki 889-2155, Japan; yuji@cc.miyazaki-u.ac.jp
[5] School of Engineering, University of Miyazaki, Miyazaki 889-2155, Japan; t0d165u@cc.miyazaki-u.ac.jp
\* Correspondence: somchai_aru@yahoo.com

**Abstract:** This study used the Area Elimination Method (AEM) of transformer design with transformer characteristic simulation. The multidimensional variable of physical parameters such as magnetic density, current density, voltage and coil turn was performed. This method was used in designing the 8000 kVA 22,000–3300/1905 V oil-type large distribution transformer. The result from a design found that the objective function tries to reduce the material to be used, and less core steel, less conductors, less transformer oil or less transformer tanks may cause an increasing load loss or increasing temperature rise, but we have calculated the temperature in the winding and design the radiator fins at the same time. After designing the transformer with AEM and it being manufactured in the production process, the transformer was tested according to IEC standards. It was confirmed that the power loss tests with no load yielded lower power loss than the standard value. In addition, the transformer passed the satisfaction test and the results of this design were built and tested with IEC 60076 standards. The information from the design using the Area Elimination Method could be a guarantee of standardized accuracy for oil distribution transformers. This also saves time and increases design efficiency for transformer designers.

**Keywords:** characterization analysis; oil-immersed transformers; area elimination technique; efficiency

## 1. Introduction

The power system can be separated into three parts: power generation system, power transmission system and distribution system. At present, most of the power generation systems are power plants such as gas power plants, thermal power plants, hydroelectric power plants and nuclear power plants. In addition, it could be a renewable energy power plant such as a solar power plant, wind power plant, wave power plant, etc. Most of the power transmission systems use an alternating current (AC) transmission system with conversion between the voltage level of sending and receiving; power plants have to step-up voltage to high voltage and connect by a transmission system to the distribution system. The distribution system receives a high voltage from the transmission line and steps-down voltage to distribute it to various consumers. In addition, the consumer requires different voltage levels such as a high voltage level or medium voltage level for the industrial

sector or low voltage for housing. Therefore, the power system requires voltage levels to be different in order to suit each type of load. The equipment that changes the AC voltage level is a transformer. Thus, it can be seen that in a power system, there are several transformers that must be used.

At present, the distribution transformer manufacturing industry has a higher competitive rate. In order to increase industrial competitiveness, manufacturers must have appropriate measures to reduce production costs in order to increase their turnover and be more competitive with other manufacturers. Therefore, in the design of distribution transformers, it must be designed at a low cost of production in order to maintain the competitiveness of the transformer manufacturing industry. The design of a transformer, due to the specifications and costs of production, is sometimes difficult due to the many parameters for the transformer's design [1,2]. In the past, transformer design was based mainly on the designer's experience with using a trial-and-error method of modifying a lot of variables as appropriate. In addition, there are many owners' specific requirements, such as maximum loss, maximum weight, width, length and height, etc., that result in it being difficult to design. Furthermore, when the designer is inexperienced, they cannot meet target characteristics and a low cost of production, in terms of the price of the main material, such as the laminate of steel core and conductor wire when imported from abroad, of which the price is uncertain. Thus, the design for the optimum production cost has to be redesigned every time when the material price changes, which makes the design difficult [3,4].

The distribution transformer's design using the original design results to develop a redesign results in a trial-and-error method according to the experience of the designers in order to obtain a new design result that is more suitable. The design results of the regularly produced distribution transformers, therefore, have a relatively reasonable design effect. On the other hand, if the design of the transformer is a new design, this results in it being difficult to calculate a suitable transformer [5,6]. Currently, there are many techniques for optimization such as Local Search (LS), Particle Swarm Optimization (PSO), Bat-Inspired Algorithm (BA), Cuckoo Search (CS) and Firefly Algorithm (FA). However, the techniques mentioned have different advantages and disadvantages. The Area Elimination Method (AE) has advantages in speed and flexibility and can be easily applied to transformer designs. Therefore, AE techniques are suitable for use in transformer design [7,8].

The AE Method is a numerical method used for searching for the optimization of conditions of an objective function. For the optimization of a transformer design in respect to structural design, we considered the effect of power loss, short-circuit impedance percentage, insulation and cooling of the winding; as a result, the structure of the distribution transformer has reasonable production costs and could actually be built. In addition, characteristics are required such as steel core, winding unit, insulation, cooling in winding unit and calculation of power loss and short-circuit impedance percentage. In addition, it also takes into account the distance factor to avoid problems from the partial discharge (PD) of the transformer when on load [9,10]. However, the values obtained for the transformer design using the AE technique are for design purposes only. Therefore, the most important aspect of the design is to use the values obtained to build the transformer and check the parameters compared to the standards, such as IEC 60076, in order to verify the correctness of the program. Thus, this research has applied AE techniques for transformer design to determine the optimization of parameters for building the transformer while keeping production costs as low as possible and testing for comparison with IEC 60076 standards [11].

## 2. Transformer Design Concept

The design of distribution transformers must have specific characteristics such as power rating, phase voltage, frequency, vector group, impedance percentage, loss power

requirements or test standards, etc. The design will consider such conditions so that the results of the method meet the objectives [12]. The procedure begins with Equation (1).

$$E_t = \frac{V_p}{N_p} = \frac{V_S}{N_S} \tag{1}$$

where the $E_t$ is the transformer ratio; $V_p$ and $V_S$ are the voltage on the primary and secondary side, respectively; $N_p$ and $N_S$ are the number of turns in the primary and secondary coil, respectively. The estimation of the initial value of Et was determined by Equation (2).

$$E_t = K\sqrt{Q} \tag{2}$$

The $K$ is the constant set between 0.3–0.6 and the $Q$ is the rated power (kVA). The $A_g$ is the core cross-sectional area which can be calculated according to Equation (3).

$$E_t = 4.44 f B_m A_g.10^{-4} \tag{3}$$

where $f$ is the power system frequency (Hz), $B_m$ is the magnetic force line density (T). When the $A_g$ is the net core cross-section area (mm$^2$) thus the diameter of the iron core ($d$) is determined from Equation (4).

$$A_g = K_1 \frac{\pi d^2}{4} \tag{4}$$

when $K_1$ is the core component coefficient, for example, $K_1$ is 0.92, 0.925, 0.93, 0.935, 0.94, 0.945, when the core layer is 6, 7, 8, 9, 10 and 11, respectively. The current density in the windings is determined by the type of conductor, level of loss and the temperature increase of the coil being industry acceptable, generally defined as no more than 3.5 A/mm$^2$. Insulation considerations shall be set to suit the test voltage rating, where the safety distance used for insulating oil designs is rated as 3 kV/mm for uninsulated oil, paper barrier, and 5 kV/mm for insulating oil, which is a specified period for industrial use. The total core weight consists of the importance of the three axes and the weights of the upper and lower yokes, which can be calculated according to Equation (5).

$$W_{Fe} = D_{Fe} A_{Fe} (3WH + 4CD + 2WD_1) \tag{5}$$

where:

$W_{Fe}$: total iron core weight (kg).
$D_{Fe}$: laminate density of $7.65 \times 10^{-6}$ kg/mm.
$WD_1$: widest laminate width (mm).
$WH$: Axial pin height (mm).
$CD$: distance between the center pin (mm).
The coil weight ($M_{cu}$) can be calculated according to Equation (6).

$$M_{cu} = D_{cu} A_{cu} L_m N \tag{6}$$

where:

$M_{cu}$: weight of the conductor coil (kg).
$D_{cu}$: conductor density (Cu = $8.9 \times 10^{-6}$ kg/mm$^3$).
$A_{cu}$: net cross-section area of conductor (mm$^2$).
$L_m$: average cycle length of the coil (mm).
$N$: number of turns of the coil.
Equation (7) shows the calculation for the no-load loss ($P_{NL}$), which can be calculated from the iron core loss value in terms of watts per kilogram (W/kg) at the magnitude of the current magnetic line density obtained from the manufacturer together [13].

$$P_{NL} = k_{wn} P_m W_{Fe} \tag{7}$$

where:

$P_{NL}$: no-load loss (W).

$k_{wn}$: multiplier coefficient from the operation and quality of the laminate.

$P_m$: power dissipation in the iron core (W/kg).

The load loss as the sum of the resistance losses in the coil and the stray losses. The power loss from the resistance in the coil can be calculated according to Equation (8).

$$R = \frac{\rho L_m N}{A_{cu}} \tag{8}$$

where:

$R$: winding resistance ($\Omega$).

$\rho$: specific resistance of the conductor winding ($\Omega$-mm).

The resistance of the winding can be found in Equation (9).

$$P_{LL} = k_{cup}I_P^2 R_P + k_{cus}I_S^2 R_S + lead_p + lead_S + stray \tag{9}$$

where:

$P_{LL}$: loss at the winding (W).

$k_{cup}$ and $k_{cus}$: operating coefficient at the primary and secondary windings, respectively.

$I_p$ and $I_s$: rated currents at primary and secondary windings (A), respectively.

$R_P$ and $R_S$: primary and secondary winding resistance ($\Omega$), respectively.

$lead_p$ and $lead_s$: losses from the primary and secondary lead wire, respectively.

$stray$: inductive loss as the effect of the inductance.

In practice, it can be estimated to be more consistent and accurate from statistical data, as in the computations in [14,15]. The reactance percentage (%X) can be calculated according to Equation (10); the resistance percentage (%R) can be found in Equation (11); and the impedance percentage (%Z) at the same length of the coil is estimated by Equation (12).

$$\%X = \left( \frac{7.91 \left( \pi f I_S N_S^2 D_m K_R \right)}{V_S H_k \times 10^7} \right) \left( a + \frac{b_1 + b_2}{3} \right) \tag{10}$$

$$\%R = \frac{P_{LL}}{Q} \times 100 \tag{11}$$

$$\%Z = \sqrt{\%R^2 + \%X^2} \tag{12}$$

where:

$V_S$: voltage per phase on the secondary side (V).

$D_m$: mean diameter at the gap between primary and secondary windings (mm).

$H_k$: mean height of the windings (mm).

$a$: gap between the secondary winding and the primary winding (mm).

$b_1$: thickness of the primary side winding unit (mm).

$b_2$: thickness of the secondary side winding (mm).

$K_R$: structural adjustment of the coil is reduced by the effect of the magnetic field as in Equation (13) and the details shown in Figure 1.

$$k_R = 1 - \left( \frac{a + T_{ws} + T_{wp}}{\pi \times l_{wm}} \right) \tag{13}$$

where:

$l_{wm}$: height of primary and secondary windings (mm).

$T_{ws}$: thickness of the secondary winding (mm).

$T_{wp}$: thickness of primary winding (mm).

These are all necessary data for using transformer design with AEM for transformer characterization with a standard transformer test [16,17]. The steps of the design is as in the next section.

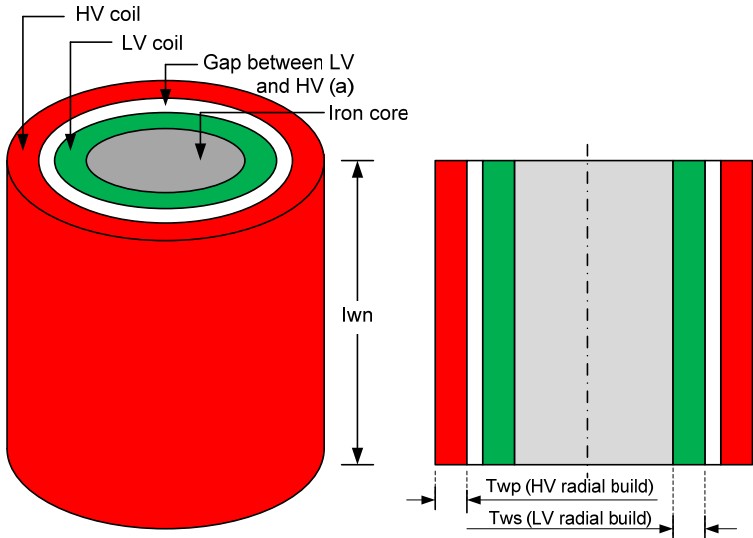

**Figure 1.** The space of winding transformer.

## 3. Step of Area Elimination Technique Program

The design transformer in this paper is to search the minimum material cost as the objective function, calculated as:

$$SB = P_{Fe}M_{Fe} + P_sM_s + P_pM_p + P_{oil}M_{oil} + P_{tk}M_{tk} \qquad (14)$$

where:

$P_{Fe}$: core steel cost in baht/kg.
$M_{Fe}$: core steel weight (kg).
$P_s$: secondary conductor cost in baht/kg.
$M_s$: secondary voltage conductor weight (kg).
$P_p$: primary conductor cost in baht/kg.
$M_p$: primary voltage conductor weight (kg).
$P_{oil}$: transformer oil cost in baht/kg.
$M_{oil}$: transformer oil weight (kg).
$P_{tk}$: transformer tank cost in baht/kg.
$M_{tk}$: transformer tank weight (kg).

The optimization method can be divided into the Direct method and Indirect method. The Direct method compares the relationships and finds the target equation to have the highest or lowest value with zero slopes. In practice, it takes a lot of time. The Indirect method compares the relationships and finds the lowest target equation for the given equation group [7,8].

For complex calculations such as transformer designs, the Indirect method can save a lot of time; in this case, the Area Elimination Technique method is an indirect method. Area Elimination Technique is suitable for dealing with multiple variable systems. To use the program to design transformers for distribution at a reasonable price, the following information must be specified [18–22].

Determine the requirements for the design results: Here, the coordinates of the distribution transformers that need to be designed for the program has to be entered, including power, primary and secondary voltage, percentage of primary voltage junction, number of phases, frequency, electrical system and vector group.

Determine the structural features: This is to enter the structural features for the program, including the type of iron core, the number of layers of the iron core, type of primary and secondary wire, type of primary and secondary coil, the minimum spacing between coils and yoke, the spacing between coils, minimum height, different insulation distances and loss of power while other loads.

Define the limitation of the condition variable: This defines the scope of variables used in the design so that the design results are in the specification area, including steel core coefficient, magnetic line density, current density and temperature rise consideration, for the specification of no-load loss, load loss and percentage short circuit impedance. This can be determined depending on the conditions of the design.

Area Elimination Method operations: This is a work schedule that will depend on the conditions of the design and duration of work. The program windows for assigning values to different variables are performed. After determining the values of various variables used in the design. The next step will go into the process of finding a suitable value for the distribution transformer structure as required. The determination of the information in this section requires some knowledge and experience of the designer.

The five steps that were used to design the transformer with AEM are as follows:

Step 1: Step of traditional industrial calculations.

Step 2: Step of changing secondary coil turn.

Step 3: Determine the middle point of finding a suitable value.

Step 4: Calculate the suitable value.

Step 5: The final adjustment to be consistent with the production.

## 4. Simulations Results

In order to begin designing a transformer with the Area Elimination Technique, the boundaries for the design must be established first. In this research, the 3-Phase distribution transformer 8000 kVA, 50 Hz, 22 kV–3300/1905 V, Dyn11 vector group and percent impedance at 7.5 has been designed under IEC 60076 standards where the distances of high voltage to Yoke, low voltage to Yoke and phase to phase was defined in millimeters. The limitations of the design parameters are shown in Table 1.

**Table 1.** Parameter design constraints.

| Parameter | Unit |
|---|---|
| $4 \times 10^2 \leq \text{kc} \leq 6 \times 10^2$ | $\text{mm}^2 \cdot \text{J}^{-1/2}$ |
| $1.0 \leq \text{B} \leq 1.75$ | T |
| $1.0 \leq \text{Jp, Js} \leq 3.5$ | $\text{A/mm}^2$ |
| $\%Z \leq 7.5$ | % |
| No-load loss $\leq 9500$ | Watt |
| Load loss $\leq 44{,}000$ | Watt |
| Gradient $\leq 18$ | °C |

However, when increasing the number of turns of the secondary winding up from 30 turns to 38 turns, the conditions for the simulation were as follows:

The constraint is the output function.

1. No-load loss
2. Load loss
3. % Impedance
4. The multidimensional variables used are:
    ○ Y(1) is the secondary number of turns.
    ○ Y(2) is the magnetic flux induction.
    ○ Y(3) is the electrical length of winding.

&#9675;  Y(4) is the secondary current density.
&#9675;  Y(5) is the primary current density.

It was observed that the cost of the transformer decreased because the laminate iron core weight, oil weight and tank and fins weight was decreased. On the other hand, it was found that the weight of the secondary winding increased due to the number of turns that were increased. However, after careful consideration, it was found that the number of secondary windings of 38 turns led to lower costs. Table 2 shows the parameters of the distribution transformer using the Area Elimination Technique in the design. Therefore, the 38 turns of the secondary winding had to be calculated as the suitable value, again, which can be shown in Figure 2, and it was symbolized as 38*.

**Table 2.** Design output.

| Item | Value |
|---|---|
| Core cross section area | 1278.9 cm$^2$ |
| Induction | 1.769 T |
| Primary winding Current density | 2.737 A/mm$^2$ |
| Secondary winding Current density | 2.737 A/mm$^2$ |
| Core weight | 5880.5 kg |
| Primary conductor weight | 1169.87 kg |
| Secondary conductor weight | 838.17 kg |
| Oil weight | 2914.6 kg |
| Tank and radiator weight | 2281.9 kg |
| No-load loss | 9056 W |
| Copper loss in primary at 75 °C | 20,541 W |
| Copper loss in secondary at 75 °C | 17,179 W |
| Other loss | 4000 W |
| Total load loss at 75 °C | 51,709 W |
| %Impedance | 7.42% |

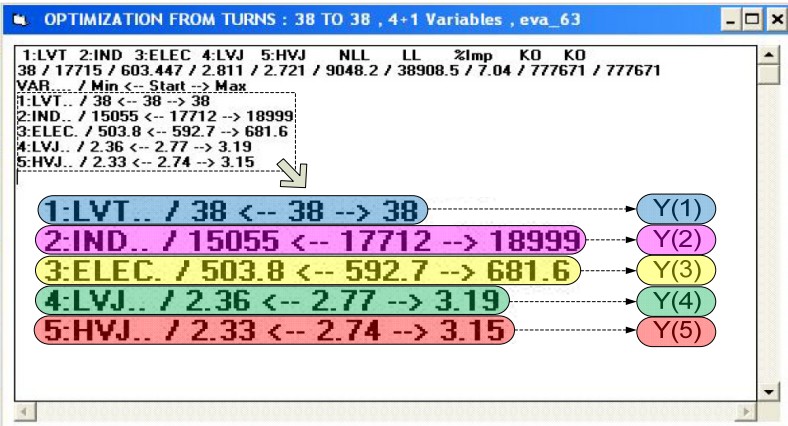

**Figure 2.** The calculation result for finding the suitable value.

Figure 2 shows the calculation results for finding suitable values:

1. The number of rounds on the secondary side changed from the number of secondary windings from 38 to 38 turns, which it was Y (1) parameter.
2. The density of magnetic force, Y (2) was between 15,055 Gauss and 18,999 Gauss.
3. Height of the coil or Y (3) parameter was between 503.8 mm and 681.6 mm.

4. Y (4) parameter, which was the current density of the secondary side, was between 2.36 A/mm$^2$ and 3.19 A/mm$^2$.
5. The current density of the primary side of the transformer or Y (5) parameter when in simulation was between 2.33 A/mm$^2$ and 3.15 A/mm$^2$.

The next step was the final adjustment for the optimization to be consistent with the production of the transformer, which was symbolized as 38**. Figures 3–8 show the required simulation results for the characterization analysis of the oil-immersed transformer obtained by the Area Elimination Method Design [23–25].

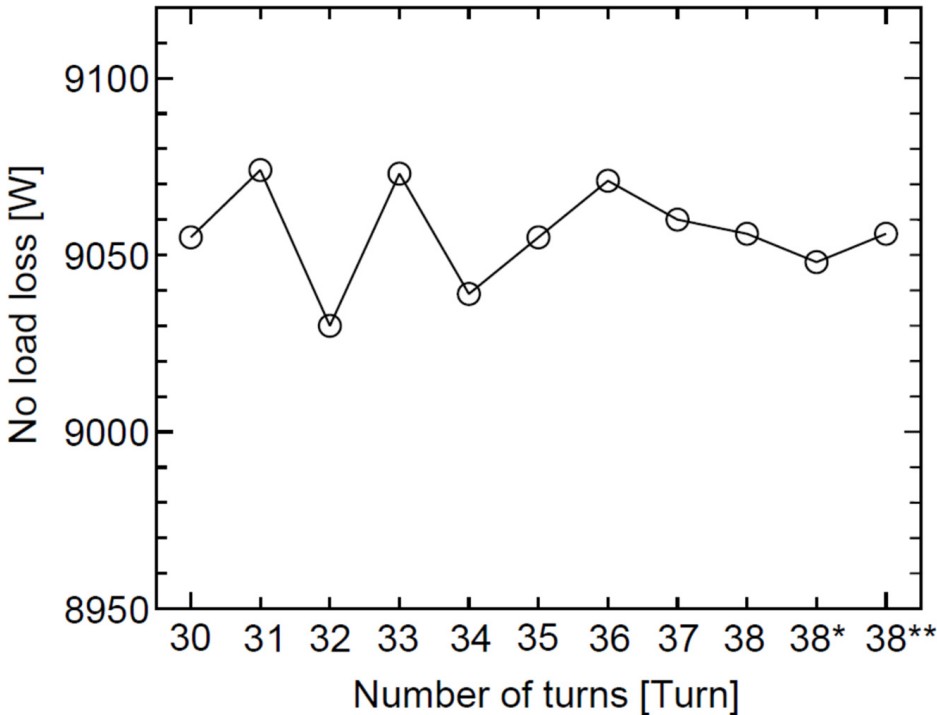

**Figure 3.** The no-load loss when the secondary winding changes.

Figure 3 shows the no-load losses when the secondary winding changes. The x-axis shows the number of turns of the secondary winding side was changed from 30 turns to 38 turns while the y-axis shows the no-load loss from the secondary winding changing. In addition, the x-axis also shows the value 38* and 38** turns with a calculation of the suitable value and the final adjustment value for the production, respectively. It was found that the maximum and minimum no-load loss was about 9074 W and 9030 W, respectively. However, it was found that the secondary winding side at 38* and 38** was about 9050 W and 9060 W, respectively.

Figure 4 shows the secondary coil loss when the secondary winding changes. The x-axis shows the number of turns of the secondary winding under the same conditions as in Figure 3, while the y-axis shows the secondary coil loss. It was found that the secondary coil loss was stable when the number of turns was 30 turns to 36 turns. In addition, it was observed that secondary coil loss tended to increase when the number of turns of the secondary winding was more than 37 turns. Thus, it was confirmed that when the number of turns of the secondary winding was 38* and 38** it resulted in the secondary coil loss being about 17,300 W and 17,600 W, respectively. The secondary coil loss was higher due to the increased number of turns on the secondary side.

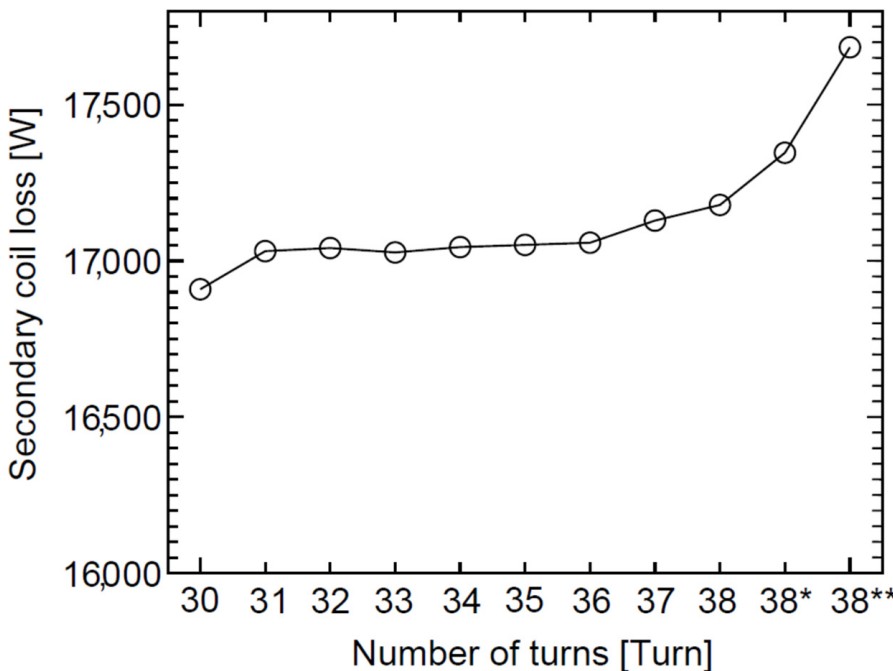

**Figure 4.** The secondary coil loss when the secondary winding changes.

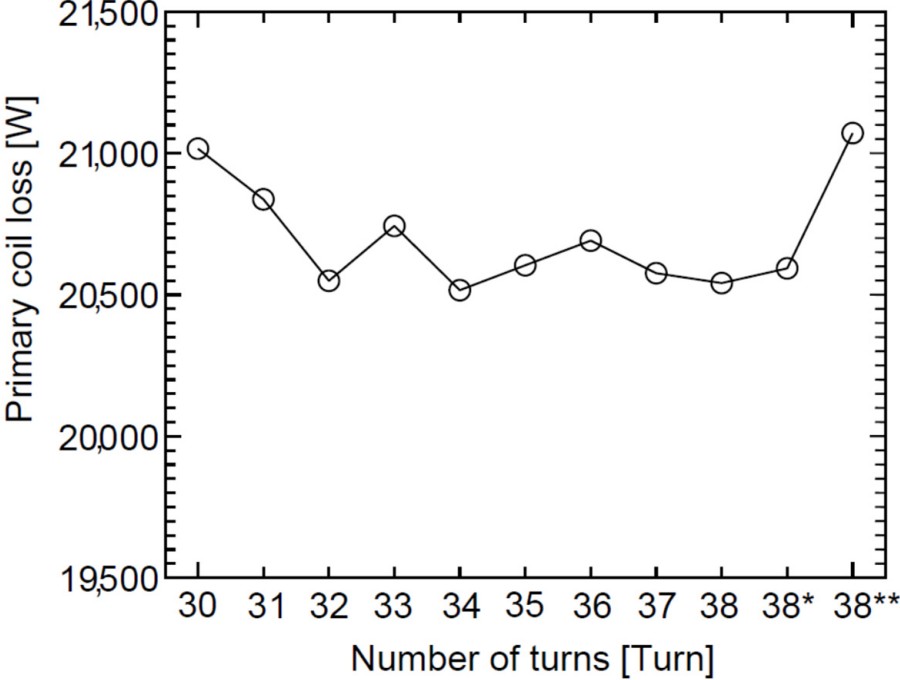

**Figure 5.** The primary coil loss when the secondary winding changes.

Figure 5 shows the primary coil loss when the secondary winding changes. The x-axis shows the number of turns of the secondary winding under the same conditions as in Figure 3, while the y-axis shows the primary coil loss. It was found that the maximum and minimum of the primary coil loss was about 21,000 W and 20,500 W, respectively. Initially, primary coil loss decreased when changing the number of secondary turns from 30 to 32. After that, it was found that the primary coil loss was stable when the number of secondary coils was 33–38* turns, in which the primary coil loss was between about 25,000–25,800 W. Finally, when the number of secondary windings changed from 38* turns to 38** turns,

the primary coil loss was increased again, which confirmed that the primary coil loss was about 21,000 W when the number of secondary windings was 38** turns.

Figure 6 shows the lead wire loss when the secondary winding changes. The x-axis shows the number of turns of the secondary winding under the same conditions as in Figure 3, while the y-axis shows the lead wire loss. It was found that the maximum and minimum lead wire losses were about 1100 W and 950 W, respectively. In addition, it was observed that the trend of the graph decreases when the number of secondary coils increases from 30 turns to 38 turns. However, it was found that the lead wire loss was increased again from conditions of calculating the suitable value and final adjustment for the production with 38* turns and 38** turns, for which the lead wire loss was about 980 W.

Figure 7 shows the total loss when the secondary winding changes. The x-axis shows the number of turns of the secondary winding under the same conditions as in Figure 3, while the y-axis shows the total loss. It was observed that the total loss was stable when the number of secondary coils was 30 turns to 38 turns, which in this period the maximum total loss and minimum total loss were about 50,900 W and 50,400 W, respectively. Nonetheless, it was found that the total loss increased when the number of secondary coils was 38* turns and 38** turns. Especially, the maximum total loss of this graph was about 51,500 W from the number of secondary coils at 38** which was a result of the summation of all losses in the transformer. However, this maximum total loss was an acceptable range for the IEC 60076 standards.

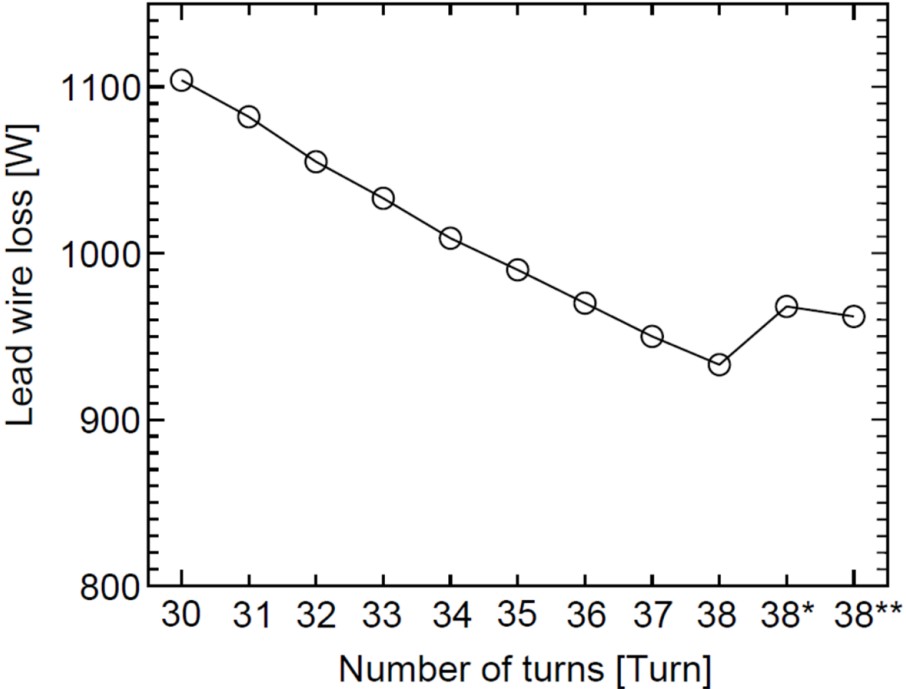

**Figure 6.** The lead wire loss when the secondary winding changes.

Figure 8 shows the percentage impedance when the secondary winding changes. The x-axis shows the number of turns of the secondary winding under the same conditions as in Figure 3, while the y-axis shows the percentage impedance. It was found that the percentage impedance was relatively stable when the number of secondary coils was between 30 turns to 38 turns, in which the percentage impedance was about 7.3–7.5%. However, the minimum percentage impedance was about 7.0% when the number of secondary coils was 38* turns. Furthermore, when the number of secondary coils was 38** turns, as a result, the percentage impedance swung back to about 7.3%.

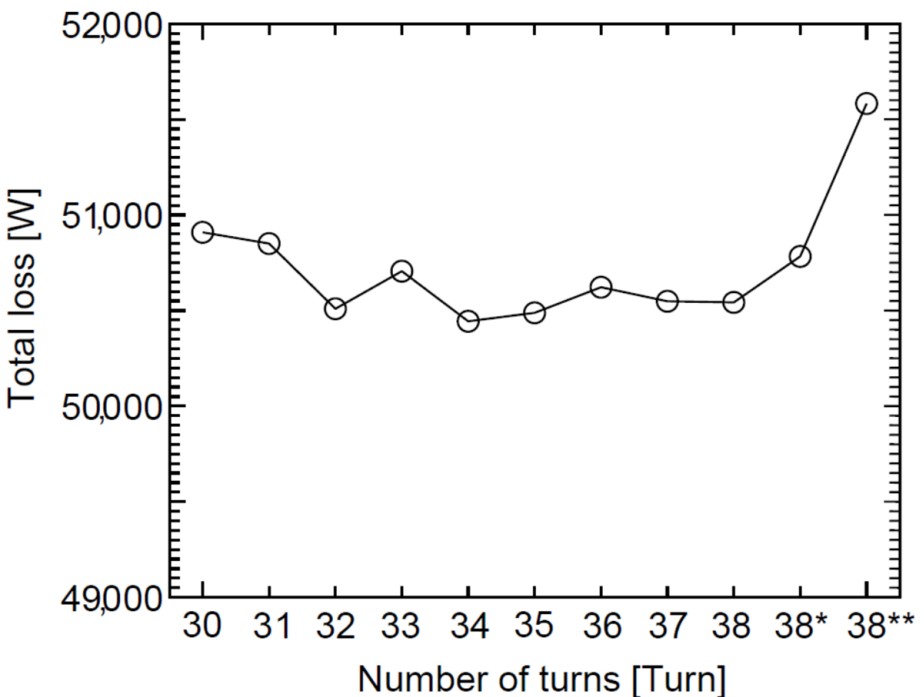

**Figure 7.** The total loss when the secondary winding changes.

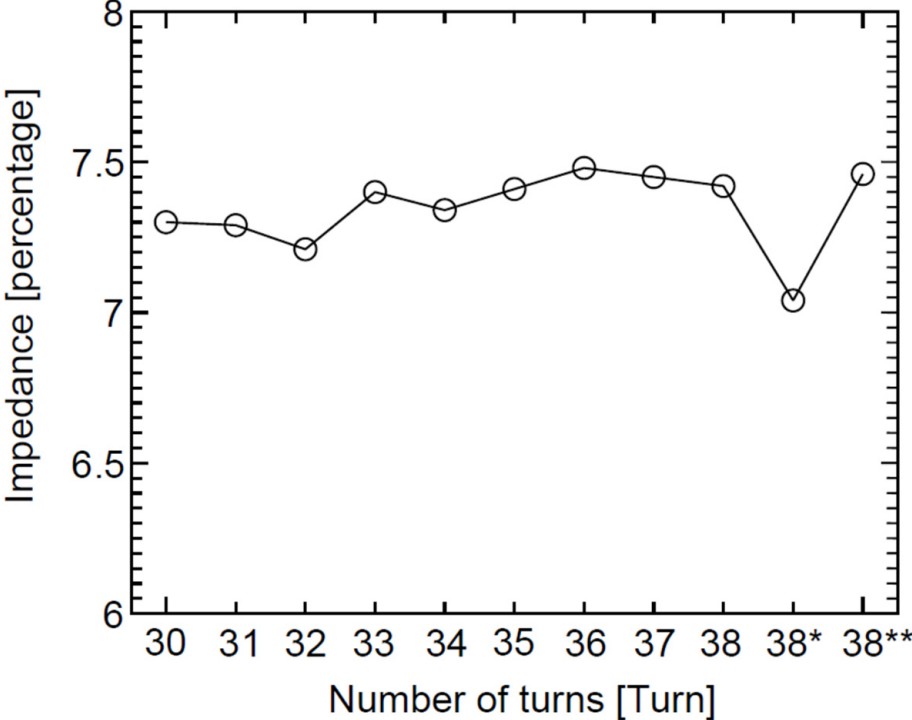

**Figure 8.** The percentage impedance when the secondary winding changes.

The results from Figures 3–8 were the design data of the number of secondary coils between 30 turns to 38** turns. It was confirmed that all design data for the 3-Phase distribution transformer 8000 kVA, 50 Hz, 22 kV–3300/1905 V, Dyn11 vector group and percent impedance at 7.5 by the Area Elimination Technique was an acceptable range for the IEC 60076 standards. Due to the number of secondary coils at 38** turns, this results in a lower cost, as already mentioned. Thus, in the manufacture of distribution transformers, it was possible to use a value of the number of secondary coils at 38** turns. Moreover,

the transformer efficiency could be calculated from Equation (15) with the percentage load at 25%, 50%, 75% and 100% under the conditions of power factor (pf) at 0.8 and 1.0, as in Figure 9.

$$\text{Efficiency} = \left( \frac{\text{Output}}{\text{Output} + \text{losses}} \right) \times 100 \tag{15}$$

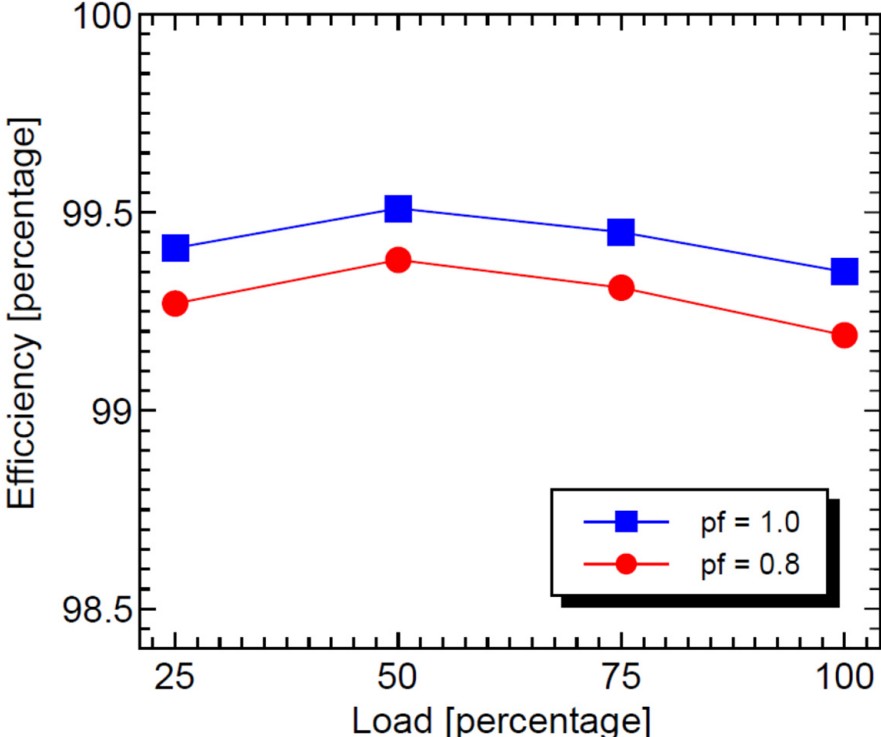

**Figure 9.** The transformer efficiency when percentage load and power factor change.

Figure 9 shows the transformer efficiency when the percentage load and power factor change. The x-axis shows percentage load at 25%, 50%, 75% and 100%, while the y-axis shows the percentage transformer efficiency, where the power factor was 0.8 and 1.0. It was found that the unity power factor was more efficient than the 0.8 power factor throughout percentage load. In addition, the efficiency at 50% of load was the highest of both the power factors of 0.8 and 1.0, which was about 99.3% and 99.5%, respectively.

## 5. Result of Testing

### 5.1. Measurement of Insulation Resistance

The insulation resistance measurement was a test to find the insulation resistance of the transformer between the high voltage winding and low voltage winding of the transformer with the ground [26–29]. The first step for the testing, the all-winding of transformer, must be short-circuits with connections to all lines of winding. After that, the winding of the transformer was connected to a Mega-ohmmeter in order to measure the insulation resistance of the transformer. The voltage for the test was set at 2500 VDC in order to supply the coil, after that we applied the voltage for about 1 min before reading the measured value; the insulation resistance test results are shown in Table 3. From the values obtained from Table 3, it was confirmed that the insulation resistance of the transformer was accepted and passed this part of the test.

**Table 3.** The insulation resistance test.

| Item | Value |
| --- | --- |
| HV-LV | 26,400 Mohm |
| HV-G | 25,600 Mohm |
| LV-G | 18,600 Mohm |

### 5.2. Over-Induced Voltage Test

Testing the over-induced voltage is a test to check the resistance to the overvoltage of windings in each phase including the various parts of the coil that are connected to the terminal. Figure 10 shows the test circuit for over-induced voltage, and the generator was supplied to the secondary winding of the testing transformer. In this testing, the digital power meter with CT and PT connections was used to measure the over-induced voltage. The frequency for testing was 200 Hz (to avoid the saturation of the iron core) at a voltage set at two times the rated voltage of the tested coil, but this must not exceed the voltage level. The test withstands the voltage according to the power frequency of other distributors, and the transformer body is grounded during the test. This test was performed on the secondary winding; therefore, the phase voltage for the test was 462 V; the test time was 30 s, following accordance with the standards.

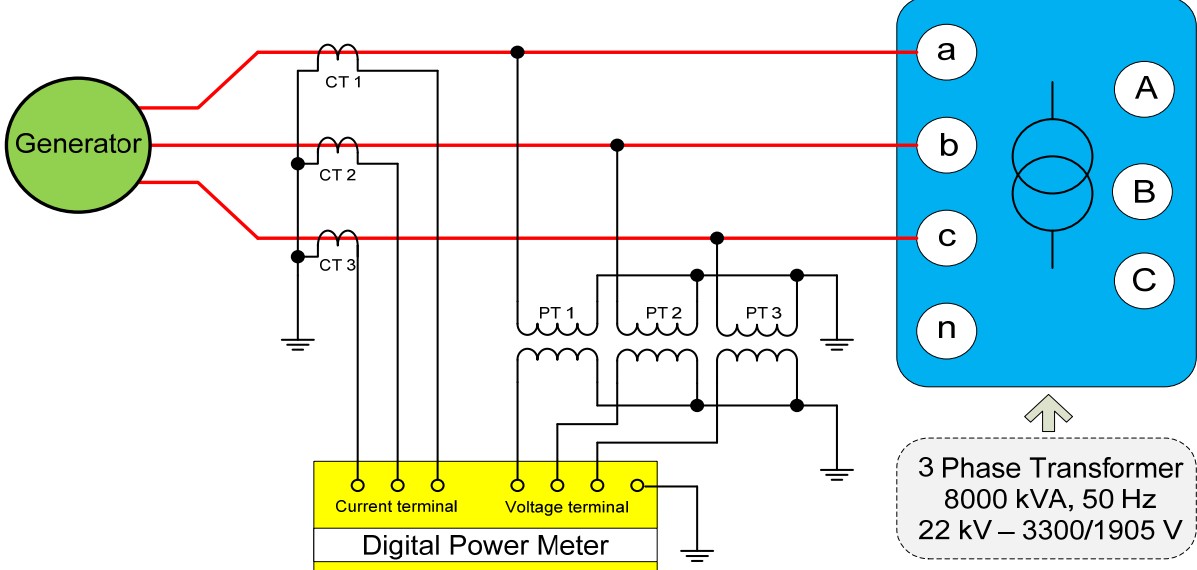

**Figure 10.** The test circuit of over-induced voltage.

### 5.3. Temperature Rise Test

The temperature rise test of the oil-type transformer was important due to the fact that oil is a liquid that could cool the transformer and keep the transformer from running at full efficiency. For the test of temperature rise, the circuit for the test was similar to the power loss test at the main junction of the transformer. The supply power was equal to the total loss of power at the 75 °C temperature of the transformer. The power supply continued until the temperature of the top oil was stable, which is determined by the temperature differences in the last 3 h (a difference not more than 1 °C). When the oil temperature is stable, it reduces the current to the rated current of the transformer for 1 h, then cuts off the current to the coil and measures the resistance of the primary and secondary windings against time. The results of resistance to time were plotted with time in order to find the resistance at the point of the cutting current ($\Omega$), then calculate the average temperature of the windings according to Equations (15) and (16), the results of which are shown in Figures 11 and 12 [15].

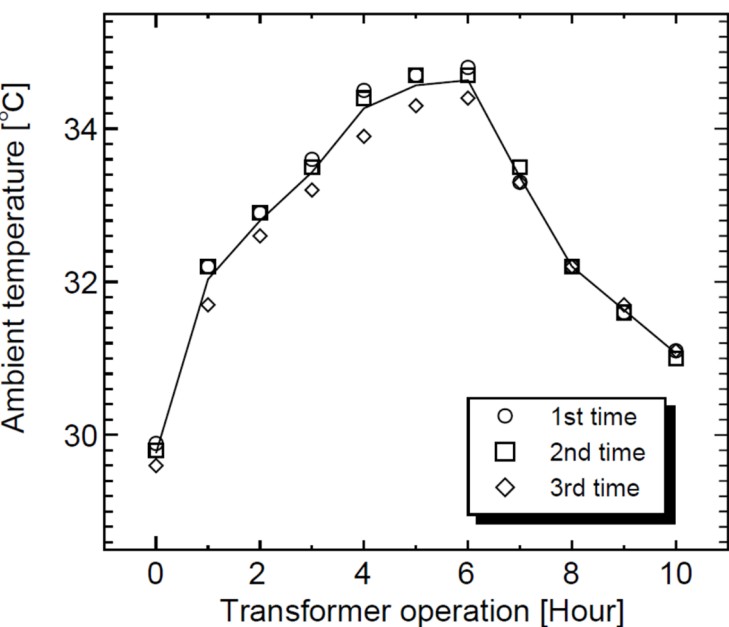

**Figure 11.** The ambient temperature during transformer operation.

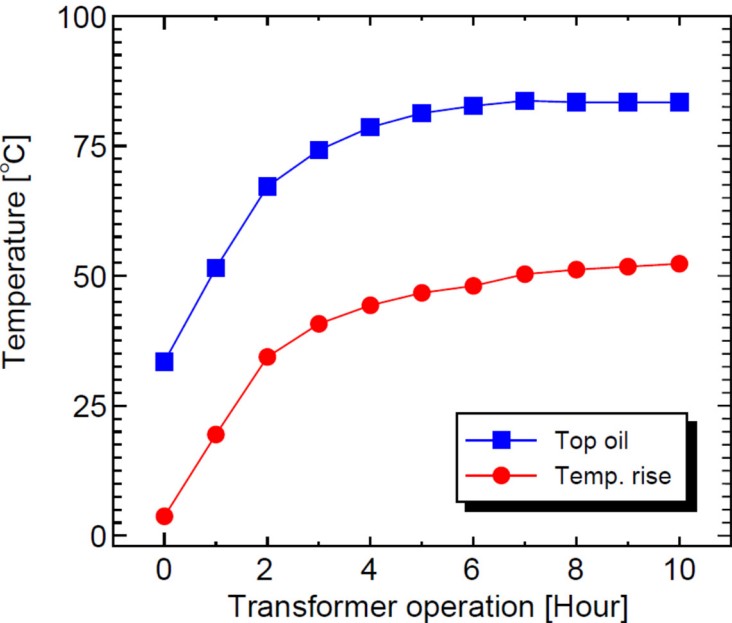

**Figure 12.** The oil temperature during transformer operation.

$$\theta_w = \frac{R_o}{R_a}(235 + \theta_a) - 235 \qquad (16)$$

$$\Delta\theta_w = \theta_w - \theta_a \qquad (17)$$

where:

$\theta_w$: average temperature of the coil (°C).

$\Delta\theta_w$: temperature rise of the coil (°C).

$R_o$: resistance value obtained from the graph at 0 min.

Figure 11 shows the ambient temperature when the transformer operation in order to find the temperature rise. The x-axis shows the operating hours of the transformer for 10 h. The y-axis shows the ambient temperature in degrees Celsius. The data were recorded three times in order to average which line in the graph was averaged from the three data. It was

found that the maximum ambient temperature was about 35 °C, caused by the operation of the transformer after the 6th hour. However, when the transformer continues to operate from the 6th hour to the 7th–10th hour, it was observed that the ambient temperature decreased due to it being during the evening time, thus reducing the temperature.

Figure 12 shows the oil temperature during the transformer operation in order to find the temperature rise. The x-axis shows the operating hours of the transformer for 10 h with the same as Figure 11. The y-axis shows the top oil temperature (blue line and mark) and temperature rise (red line and mark). It was observed that both top oil and temperature rise were trending to increase when transformer operation increased. The top oil was measured and recorded each hour. However, the temperature rise was a subtraction between the top oil and the average ambient temperature. It was observed that the temperature of the top oil was stable in the 9th hour. In addition, the top oil temperature was about 83 °C. The average ambient temperature was about 31 °C. It could be calculated that the temperature rises to the top oil by about 52 °C.

Figure 13 shows the loss comparison between the IEC 60076 standard and transformer test result, which consists of no-load loss, load loss and total loss where the orange bar is the IEC 60076 standard and the blue bar is the test result. It was found that the standard value was greater than the test value for all loss items. It was confirmed that the transformer design by the Area Elimination Technique made it possible to create a transformer due to the results of the loss test being less than the standard values. In addition, another important variable was temperature, which is shown in Figure 14.

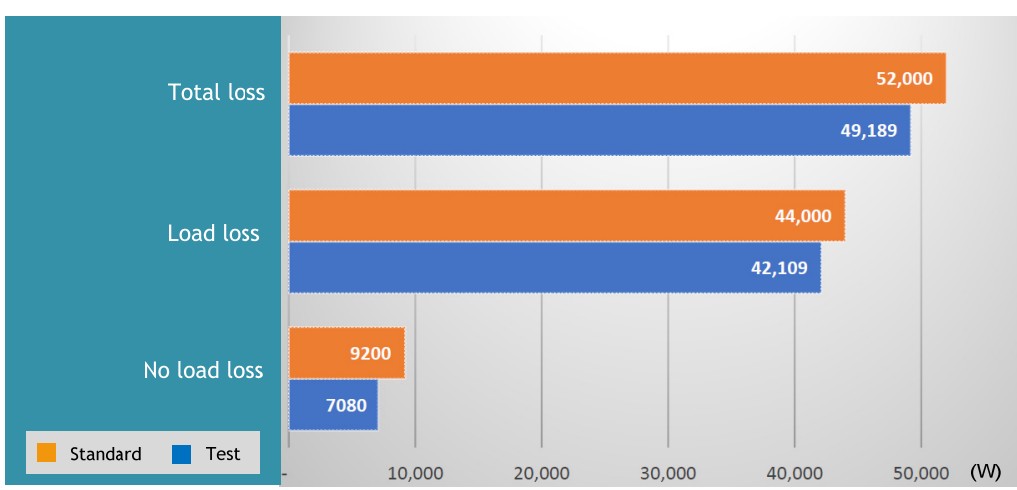

**Figure 13.** The loss comparison between IEC 60076 standard and transformer test result.

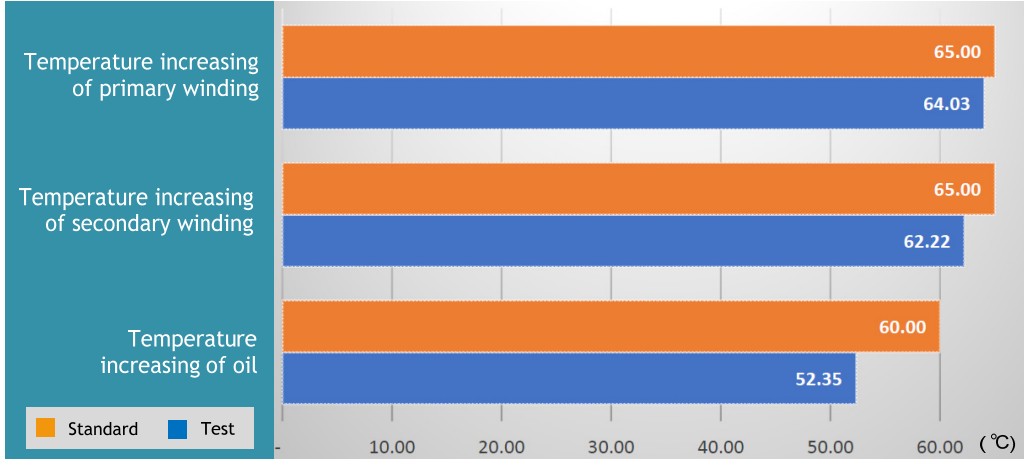

**Figure 14.** The temperature comparison between IEC 60076 standard and transformer test result.

Figure 14 shows the temperature comparison between the IEC 60076 standard and transformer test result, in which temperature consists of temperature increasing of primary winding, temperature increasing of secondary winding and temperature increasing of oil where the color of the bar is under the same conditions as in Figure 13. It was found that the standard value was greater than the test value at all loss items. Therefore, it can be assured that the design output by the Area Elimination Technique did not exceed the value specified by the design or standard results [11–13]. In addition, it can also be used to create a transformer that can reduce costs.

The main cost of raw materials for transformer production was the iron core, copper winding, insulation oil and tank and fin of the transformer, the cost of which was priced in THB. Figure 15 shows the cost comparison between the original design and the Area Elimination method. The difference between the original design and the Area Elimination method for the number of turns was 30 turns and 38 turns, respectively. It was found that the cost of the original transformer design was lower than the Area Elimination method using copper winding and tank and fin of transformer, whereas the cost of the Area Elimination method was lower than the original transformer design using iron core and insulation oil. However, the cost of the copper winding, insulation oil and tank and fin of transformer has a cost of about THB 80,000–120,000, which are very close between the original design and Area Elimination method. On the other hand, the difference mainly depends on the iron core. It was observed that the price between the Area Elimination method and the original design was about THB 400,000 and 600,000, respectively. Thus, it was confirmed that the cost of Area Elimination method was lower than the original design when summarized of all costs.

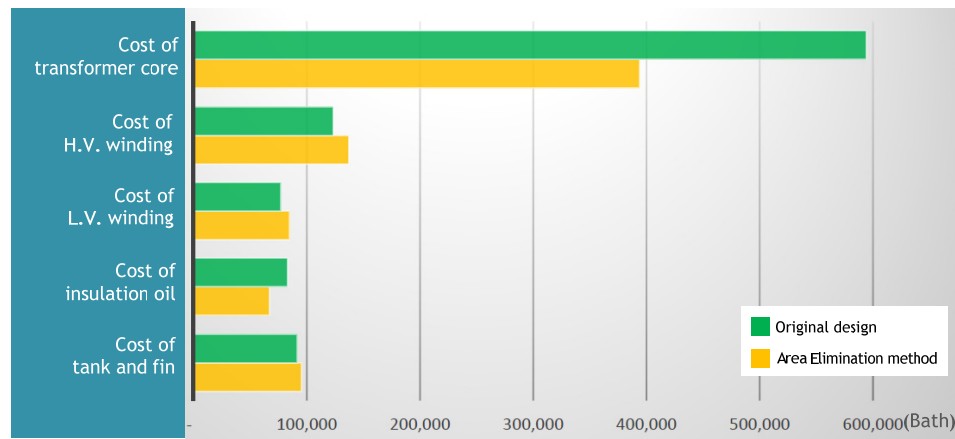

**Figure 15.** The cost comparison between the original design and Area Elimination method.

## 6. Conclusions

This research has been very successful with the characterization analysis of the oil-immersed transformers obtained by the Area Elimination Method Design. Another important factor for the design by the Area Elimination Method was that the time period used for the calculation was a lot less time, and it could also be repeated as a result of the minimal materials of the transformer. In order to design the distribution transformers, the results from the Area Elimination Method were used to create transformers with reasonable prices. The results from the Area Elimination Method were used to create real prototypes to verify the accuracy of the program. When the program designed for distribution transformers is accurate, it can be analyzed to design transformers for distribution at a reasonable price in various ways as required. The result from a design found that the objective function tries to reduce the material to be used, and less core steel, less conductors, less transformer oil or less transformer tanks may cause an increasing load loss or increasing temperature rise. However, after the real manufacturing of the transformer was finished, the transformer passed the test and followed the IEC 60076 standards with satisfaction. Thus, the

information from the design using the Area Elimination Method could be a guarantee of standardized accuracy for oil distribution transformers.

In addition, the Area Elimination Method could reduce costs when coil winding was changed from 30 turns to 38 turns. It was seen that the number of turns of the winding increased, and as a result, the cost of windings increased. The cost of low voltage winding and high voltage winding increased to 9.6% and 11.2%, respectively. In addition, the cost of transformer tanks and fins slightly increased by only 4%. On the other hand, when the number of turns changed to 38 turns, this resulted in reducing the insulation oil of transformer to 19.5% because the volume of the transformer tank for oil was smaller. However, the thing that affected the cost the most was the iron core, which could be cost reduced to 33.7% at most. Therefore, the total cost when using the Area Elimination Method for design could be reduced to 19.7% when comparing with the original transformer design. In the production process, reduced costs mean increased profits. In addition, the Area Elimination Method can save time for design and increases design efficiency in order to produce professional designs. In the further study, the electromagnetic distribution by FEM will be performed with a commercial professional program.

**Author Contributions:** Conceptualization, N.T. and S.A.; methodology, N.M. and K.T.; validation, A.S. and K.T.; formal analysis, N.T., A.C. and T.T. (Thaweesak Tanaram); investigation, T.Y. and H.K.; writing—original draft preparation, W.P. and C.C.; writing—review and editing, N.T.; visualization, N.T. and T.T. (Tanes Tanitteerapan); supervision, N.T. and N.M.; project administration, S.A. and K.T. All authors have read and agreed to the published version of the manuscript.

**Funding:** This work was supported by King Mongkut's University of Technology Thonburi (KMUTT), Thailand, and under the project of the Research, Innovation, and Partnerships Office (RIPO) with the Faculty of Industrial Education and Technology Research Funding.

**Institutional Review Board Statement:** Not applicable.

**Informed Consent Statement:** Not applicable.

**Data Availability Statement:** The data presented in this study are available on request from the corresponding author.

**Acknowledgments:** The authors would like to thank King Mongkut's University of Technology Thonburi (KMUTT), Thailand, for their support to the laboratory and research facility.

**Conflicts of Interest:** The authors declare no conflict of interest.

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
