# Peer review of "The Characterization Analysis of the Oil-Immersed Transformers Obtained by Area Elimination Method Design"

_applsci, doi:10.3390/app12083970_

Round 1
Reviewer 1 Report
The conclusion should be re-written. It should reflect the outcome of the study without discussion and there should not be any citation in this section. Also, the further scope of the study should be mentioned.
Author Response
Dear Sir
We make an answer as an attached file.
Best Regards
Mungkung N

Reviewer 2 Report
The authors present a new method of designing transformers, which greatly simplifies its design and calculation, without loss of efficiency. A review has been made on the current topic, in which, however, not many works have been considered, I would like to expand this part. The following is a calculation of the proposed solution. I would also like to note that the title of section 2 contains a link, which would be better not to do. All the necessary experiments were carried out to confirm the idea and a lot of necessary characteristics were taken, to which there are no complaints. I would also like to expand the conclusion section, add quantitative results there, how many times this or that characteristic has been improved in comparison with other solutions. In addition, the part after the conclusion was not completed, in particular, what part of the work was performed by each of the authors, etc. The work undoubtedly deserves acceptance, but after revision. Yes, I completely forgot in the conclusion there is a proposal of a rather formal content, it is better to avoid such constructions. "but we have calculated" line 553.
Author Response
Dear Sir
We make an answer as an attached file.
Best Regards

Reviewer 3 Report
- In equation (3) was used the vector multiplication sign, i.e. "x". This is a scalar multiplication so it should be a dot. Similarly, for example on lines 135, 145. Please check the full article and correct it.
- In the caption at Fig. 1, the inscription must start with a capital letter.
- Sections 3.1-3.4 each contain one paragraph. I believe that in this case it is not necessary to break this content into subsections.
- "Y" was missing from line 302.
- Throughout the article, there are real numbers written with a point or a comma and even in two forms, ie 1.169.87 kg. It is necessary to standardize because it is not known what is in the decimal.
- The numbering of chapters and subsections is incorrect as "5.2" appears twice.
- I believe that authors should carry out a more careful review of the literature and clearly emphasize their contribution to the science and the novelties they propose.
- In Conclusions lacks a clear emphasis on the advantages and disadvantages of the proposed solution and the possibility of its modification.
Author Response

(The authors gave the same response as above.)

Round 2
Reviewer 3 Report
I have no more questions. I am just asking you to correct the equation (14), because in my opinion it should contain a percent sign.
Author Response
Dear Sir
In your opinion, equation (14) should have a percent sign.
That is correct as an amount of the budget cost.
Best Ragards
Mungkung N
